# Awareness of climate change's impacts and motivation to adapt are not enough to drive action: A look of Puerto Rican farmers after Hurricane Maria

Luis Alexis Rodríguez-Cruz[1,2]*, Meredith T. Niles[2,3]

**1** Food Systems Graduate Program, University of Vermont, Burlington, Vermont, United States of America,
**2** Gund Institute for the Environment, University of Vermont, Burlington, Vermont, United States of America,
**3** Department of Nutrition and Food Sciences and Food Systems Graduate Program, University of Vermont, Burlington, Vermont, United States of America

* lrodrig2@uvm.edu

**Data Availability Statement:** Data cannot be shared publicly because only a portion of the data was used in the reported study. the de-identified data used in this paper can be accessed at the

## Abstract

Understanding how perceptions around motivation, capacity, and climate change's impacts relate to the adoption of adaptation practices in light of experiences with extreme weather events is important in assessing farmers' adaptive capacity. However, very little of this work has occurred in islands, which may have different vulnerabilities and capacities for adaptation. Data of surveyed farmers throughout Puerto Rico after Hurricane Maria (n = 405, 87% response rate) were used in a structural equation model to explore the extent to which their adoption of agricultural practices and management strategies was driven by perceptions of motivation, vulnerability, and capacity as a function of their psychological distance of climate change. Our results show that half of farmers did not adopt any practice or strategy, even though the majority perceived themselves capable and motivated to adapt to climate change, and understood their farms to be vulnerable to future extreme events. Furthermore, adoption was neither linked to these adaptation perceptions, nor to their psychological distance of climate change, which we found to be both near and far. Puerto Rican farmers' showed a broad awareness of climate change's impacts both locally and globally in different dimensions (temporal, spatial, and social), and climate distance was not linked to reported damages from Hurricane Maria or to previous extreme weather events. These results suggest that we may be reaching a tipping point for extreme events as a driver for climate belief and action, especially in places where there is a high level of climate change awareness and continued experience of compounded impacts. Further, high perceived capacity and motivation are not linked to actual adaptation behaviors, suggesting that broadening adaptation analyses beyond individual perceptions and capacities as drivers of climate adaptation may give us a better understanding of the determinants to strengthen farmers' adaptive capacity.

following: Rodríguez-Cruz, Luis Alexis; Meredith T. Niles, 2020, "Puerto Rican farmers' climate change and adaptation perceptions after Hurricane Maria", https://doi.org/10.7910/DVN/SO3ILC, Harvard Dataverse, V1, UNF:6:LvbAL1TsaSsfs0Op3Fxi9g== [fileUNF] Retrived from: https://dataverse.harvard.edu/dataset.xhtml?persistentId=doi:10.7910/DVN/SO3ILC.

**Funding:** MTN allocated funding from this study, which came from the College of Agriculture and Life Science and the Food Systems Graduate Program of the University of Vermont. The funders had no role in study design, data collection and analysis, decision to publish, or preparation of the manuscript.

**Competing interests:** I have read the journal's policy and the authors of this manuscript have the following competing interests: MTN is a member of the board of directors of The Public Library of Science (PLOS). This role has in no way influenced the outcome or development of this work or the peer review process, nor does it alter our adherence to PLOS ONE policies on sharing data and materials.

# Introduction

Exposure, sensitivity, and adaptive capacity are three main determinants of farmers' vulnerability to natural hazards [1, 2]. Farmers in small island states and territories, most of which are located in the tropics, farm in regions that are often disproportionately exposed and sensitive to natural hazards [2–4]. Exposure results from islands being situated in regions highly prone to extreme weather events (e.g. Atlantic's 'Hurricane Belt'), and sensitivity relates to the degree to which such hazards change physical systems (e.g. farms) [1, 4–6]. As such, strengthening adaptive capacity—defined as the set of actual abilities and resources individuals and populations have to anticipate, withstand, cope with, and recover from a hazard, and the potential or abilities they have to modify the system in order to be more resistant to impacts [3, 7, 8]—is key in decreasing vulnerability to natural hazards.

Adaptive capacity is complex, and involves many diverse and interrelated determinants, which expand from individual to social and political scales [3, 4]. It should not be confused with perceived self-capacity, which refers to people's beliefs around their own capabilities to undergo a change or carry out an action [9–11]. In the case of farmers, the adoption of new adaptation practices, technologies, and strategies, as well as the access to institutional resources and networks of support, are important for strengthening adaptive capacity [12–16]. Research in the Caribbean and Central America—areas exposed to Atlantic hurricanes—have shown that farms and farmers' characteristics, such as farm size, levels of education and income, and production styles, as well as their access to diverse markets, and sources of support, are important in reducing vulnerability to natural hazards [12, 17–20]. Nevertheless, given the heterogeneity of the regions' social and political systems, generalization of results is complicated; hence, place-specific research is important to better understand adaptive capacity [4, 21].

This is evident in the Caribbean, given the region's neocolonial dynamics, where many islands' sovereignty resides on continental countries [22–24]. This is the case of Puerto Rico as an unincorporated territory of the United States. Studying Puerto Rican farmers' climate change adaptation—the response to climate change-related impacts through the adoption of management mechanisms—could allow us to understand the extent to which biophysical, social, and political components of a system impact farmers' decision-making. This then may improve our knowledge to strengthen adaptive capacity in the Caribbean and beyond.

Perceptions are an important factor to explore in adaptation decisions [3, 6, 25], especially risk, perceived-capacity, and motivation to adapt, which have been found to be linked to the adoption of adaptation strategies [9, 26, 27]. While climate change adaptation requires external resources (extrinsic), individual determinants (intrinsic) also allow better understanding of adaptive capacity [5, 6, 14]. Studies have found that individuals' perceived capacity, perceived vulnerability, motivation to adapt, and other social and cognitive variables are key in understanding adaptation behaviors [9, 13, 25, 26]. The ways farmers perceived their capacity to adopt new adaptation behaviors, and the risks they are exposed to have been shown to drive the adoption of new agricultural practices and pro-environmental behaviors [13, 26, 28–30]. While existing research demonstrates a link between climate perceptions, beliefs, and climate change adaptation amongst the general public [e.g. 31], and farmers [e.g. 26], there are varied results in the strength to which those perceptions drive change [9, 32, 33]. Importantly, most research focused on perceptions about climate, and their role in adaptation is from high-income countries [9]. There is limited evidence of this topic from small island developing states and territories, whose unique exposure and sensitivity to climate change and extreme events may influence how residents perceive climate change and their capacity to adapt to it.

To address the current gaps in the literature around the role of perceptions in the adoption of adaptation strategies, we examine survey results from 405 Puerto Rican farmers following

Hurricane Maria, a category four hurricane, which impacted Puerto Rico in September 2017. We examine the extent to which Puerto Rican farmers' adoption of agricultural practices to prepare for future events after Hurricane Maria relates to their perceptions of climate change, self-capacity, and motivation to adapt. We draw our theoretical grounding from Spence and colleagues by analyzing the data through the lens of the psychological distance of climate change—simply put, how near or far people perceive climate change to be from themselves in different dimensions [31, 34].

## Theoretical framework

Construal Level Theory states that humans imagine the past and future through abstract constructs, since we are embedded in the here and now [35, 36]. Psychological distance, a component of the aforementioned theory, is a subjective experience that uses the self as a reference point [35, 36]. Psychological distance is measured in four components: temporal, social, spatial, and hypothetical (uncertainty of the distance or issue). The more distant we perceive an issue or event to be across these dimensions, the more we rely on abstract constructs to make sense of it. In contrast, things we perceive as closer are more concrete. For example, as an event gets closer in time (e.g. conference presentation), the awareness of the dynamics involved become more concrete (e.g. transportation to the venue, nervousness, time management, and other variables). Thus, Construal Level Theory explains that we use mental constructs to build understanding of what transcends the here and now.

Applied in the realm of climate change communication and adaptation research, the psychological distance of climate change [34] refers to how near or far people (e.g. farmers) perceive climate change to be from themselves across the four dimensions: temporal (when climate change will occur or if is occurring), spatial (where it happens), social (to whom it happens), and the hypothetical or uncertainty dimension (certainty of whether it occurs or not). Theoretically, this posits that individuals who are psychologically close to climate change are prone to perceive higher risks to hazards, and a greater motivation to act to adapt or mitigate climate change. Nonetheless, this assumption has not been consistent in research [32, 33].

## Research on the psychological distance of climate change

The aforementioned hypothesis has been the basis for many existing studies on the psychological distance of climate change (e.g. understanding how psychological distance affects the propensity for individuals to take action to mitigate or adapt to climate change). The current body of research focused on the psychological distance of climate change is both experimental [37, 38], and observational [34, 39]. More specifically, scholarship has explored (1) the interrelatedness of the different dimensions, (2) the role of experiencing extreme weather events on individuals' psychological distance of climate change, and (3) the degree to which psychological distance of climate change affects behavior (intention and actual change). Overall, evidence indicates inconsistent outcomes related to the psychological distance of climate change and its capacity to prompt action, as well as the understanding of how experiences with extreme events affects psychological distance, and the relatedness of the four dimensions varies [33].

These results may suggest that place matters; a natural hazard (e.g. hurricane), for example, can be experienced and perceived differently depending on a populations or individuals' attributes (e.g. sociodemographic characteristics, beliefs, adaptive capacity) [1, 2].

Some studies support that the four dimensions of the psychological distance of climate change are interrelated. Moreover, studies suggest that being psychologically close impacts behavior. For example, Spence and colleagues (2011, 2012) demonstrated that reported experience with floods was linked to individuals' perceptions of climate change in the United

Kingdom. Their studies showed that the four dimensions of the psychological distance of climate change were interrelated, and that reduced psychological distance related to prompting climate change mitigation actions amongst individuals [31, 34]. Furthermore, their study showed that direct experience with flooding was linked to higher perceived self-capacity [31].

Similarly, but based in the United States, Singh et al. (2017) found that the four dimensions were interrelated, and that those psychologically close expressed more support towards climate change adaptation policies. In that study, psychological distance was linked to experience with extreme events. The temporal and uncertainty dimensions, by their own, did not have significant effect on policy support; however, those that expressed higher uncertainty over climate change and were socially distant, expressed less support towards policies [39]. Furthermore, research has shown that the dimensions, though often interrelated, depending on different social and geographical attributes, manifest differently [33]. Focused on New Zealand's farmers, Niles et al. (2015) applied the psychological distance of climate change framework in combination with Liebig's Law of the Minimum to assess adaptation behaviors, finding that decreasing psychological distance may prompt action from farmers. Farmers who perceived climate change impacts as spatially close, were more likely to feel more concern for climate change and motivation to adapt [26].

Research has also described how experience with extreme weather events relates to being psychologically close [26, 31, 33, 39]. Acharibasam and Anuga (2018) found that farmers who had experienced extreme events were socially and spatially close, and that psychological distance mediated emotional regulation among farmers—the mental processes through which individuals decide how emotions are expressed and experienced [40]. Nevertheless, a recent meta-analysis performed by van Valkengoed and colleagues (2019) found that experience had small-to-moderate effects on climate change adaptation. That same meta-analysis found that perceived self-capacity had higher effects on adaptation behaviors than experience. Larcom (2019) found that experience with a heatwave amongst the general population did not elicit adaptation behaviors in the United Kingdom, but did elevate concern [41]. Similarly, Albright and Crow (2019) found that direct flooding experience was not a predictor of concern for climate change amongst Colorado, United States, residents [42]. These more recent studies suggest that there may be a threshold of individual exposure to natural hazards and their potential to catalyze behavioral change; indeed, as the rate of extreme events increases, it could be probable that such events may saturate in people's experiences and become "more normal", part of the status-quo.

The aforementioned studies also highlight the importance of taking into account how different components of a system interconnect with individual attributes, but also bring attention to how psychological distance of climate change research is limited in its generalizability [32]. Furthermore, most research on the matter has used data on reported experience, and not on direct experience (e.g. material impacts such as damage). Zanocco et al. (2018) performed case studies of residents whose communities were directly impacted by extreme events, such as tornados and wildfires, or were proximate to where the events occurred, in order to understand the relationship between experience of extreme event and views of climate change. Applying Construal Level Theory perspectives of psychological distance, they found that spatial proximity to the event did not align with climate change views, but reported harm did [43]. This quote from Demski et al. (2017), as cited by Zanocco et al. (2018), underscores the aforementioned: "Self-reported measures of experience that include: direct physical experience and material impacts of an event, well-defined in terms of personal effects and damage, are less susceptible to biased reporting, providing the best proxy available for objective experience" (p. 152) [43].

This paper attempts to fill several gaps in the research. First, we will assess Puerto Rican farmers' psychological distance of climate change after experiencing an extreme weather

event. Hurricane Maria affected all of Puerto Rico in September 2017, causing significant damages to infrastructure, agriculture, and livelihoods. The 2017 Atlantic Hurricane Season is noted for being a devastating one in the Caribbean region [44]. To the best of our understanding, there are no studies exploring the psychological distance of climate change amongst farmers in small island developing states and territories after an extreme weather event. Second, we will assess the extent to which both reported experience with other similar extreme events, and direct damages reported by Hurricane Maria relate to psychological distance of climate change. As stated above, there are mixed results on how reported experience and actual experienced damages are linked to climate change perceptions. Finally, drawing from Spence et al. (2011) and others who have shown that perceived capacity is linked to adaptation, we examine farmers adoption of agricultural practices after Maria, and its relationship to climate change distance, vulnerability, perceived capacity, and motivation to adapt.

## The present study

This study focuses in Puerto Rico, an unincorporated territory of the United States that imports around 85% of the food it consumes, and that is going through a social-economic crisis because of its billion dollar debt and political situation [44–47]. In response to Puerto Rico's debt crisis in 2016, the United States' government created a Fiscal Management and Oversight Board that, alongside the local government, imposed austerity measures, and made visible Puerto Rico's lack of political agency as a territory of the United States [22, 23]. Simultaneous and previous to this change, the agricultural sector in Puerto Rico was experiencing positive changes, such as a rise in new farmers, production increases, and increased awareness of food security [17, 48–50]. Such agricultural gains were important since Puerto Rico, like the broader Caribbean, was experiencing a steady decline in farms since the 1990's due to trade liberalization, globalization, and other external and local forces [18, 49, 50, 51, 52]. Nevertheless, such gains were erased when both Hurricane Irma and Hurricane Maria, two of the strongest in the Atlantic's history, hit Puerto Rico in September 2017 [46, 53–55].

Puerto Rico's Department of Agriculture (2018) reported that both hurricanes caused $2 billion in total losses, with the majority of losses from Maria ($228 million in production losses, and $1.8 billion in agricultural infrastructure losses). Such damages significantly impacted Puerto Rican agriculture, which, according to the USDA, is comprised of many small-scale farmers with an average farm size of 59 acres (approximately 23 hectares), who focus mostly on domestic markets, and who have an average annual income of less than $20,000 [56]. Focusing this study on Puerto Rican farmers after experiencing a major storm, not only allow us to advance our understanding of the role of direct hazard experience on the psychological distance of climate change, but provides us the opportunity to produce important applied knowledge around strengthening adaptive capacity through assessing climate change adaptation at the individual level [26].

## Objective, research questions, and hypotheses

Our study intends to assess (1) the psychological distance of climate change amongst farmers in Puerto Rico, an unincorporated territory of the United States, after an extreme-weather event, (2) the extent to which psychological distance of climate change is related to reported experience, and reported damages caused by Maria, and to (3) other perceptions important in adaptation research, such as motivation to adapt, perceived capacity (self-efficacy or perceived instrumentality), and perceived vulnerability, and (4) the role of these perceptions in the adoption of agricultural practices after Maria as a function of farmers' psychological distance of climate change.

**Fig 1. Hypothesized model.** Farm and farmer characteristics' variables included as controls.

We ask: 1) How do farm and farmer characteristics, and experiences with extreme events (i.e. reported damages and reported experience with similar events) relate to the psychological distance of climate change? 2) In light of experiencing an extreme event, what is the role of psychological distance of climate change in motivating Puerto Rican farmers to adapt to climate change? And, 3) to what extent do adaptation perceptions relate to actual adoption of agricultural practices and management strategies after Maria? We hypothesize that farmers' psychological distance of climate change will be driven by their experience (e.g. reported damages) with Hurricane Maria and past extreme weather events (H1). Furthermore, we predict that psychological distance of climate change will be related to perceived capacity, perceived vulnerability, and motivation to adapt (H2), and that these will be positively linked to the adoption of agricultural practices and management strategies after the hurricane (H3). Fig 1 shows our hypothesized model.

## Methodology

### Data gathering

Previous studies of on-farm management and climate change perceptions, and on the psychological distance of climate change [16, 26, 31, 39, 57] informed the development of a survey instrument, which was modified to fit the general objectives of the main study. This study was carried out in collaboration with the Extension Service of the University of Puerto Rico at Mayagüez. Thus, feedback from Extension partners were also included in the survey's development. The Committees on Human Subjects Serving the University of Vermont and the UVM Medical Center at the Research Protections Office approved our study on December 2017. Consent from participants was obtained orally, and was included in the survey booklet. The survey was translated into Spanish, and piloted in February 2018 with a pool of Puerto Rican farmers (n = 32), and was also shared amongst Extension agricultural agents for feedback—enumerators of the survey. Minimal language and structural changes were made to the instrument.

Surveys (n = 440) were randomly administered to farmers by Extension agents, according to the five regions the Extension Service divides Puerto Rico: Arecibo, Caguas, Mayagüez, Ponce, and San Juan. Each region has several agents that provide service to the municipalities that comprise each region. Survey deployment was based on Extension Services' recommendations to access a wide range of farmers, and because agricultural agents had an established presence. Each regional office received a set of surveys; San Juan, Caguas, and Ponce received 100 surveys each, and Mayagüez and Arecibo received 75 surveys each. Participant Extension agents randomly surveyed farmers in the municipalities they worked in. Surveys were administered between May and July 2018 in person. Farmers from 65 of Puerto Rico's 78 municipalities answered the survey. A total of 405 surveys were completed, resulting in an 87% response rate based on the American Association for Public Opinion Response Rate Calculator [58].

## Assessing the psychological distance of climate change

Eight questions were used to assess the four dimensions of the psychological distance of climate change (temporal, social, spatial, and hypothetical/uncertainty) (Table 1). These questions were adapted from previous research on the topic, and contextualized to Puerto Rico [26, 31, 39, 57]. The temporal dimension was assessed with one item, while the social and

**Table 1. Survey variables used in this study.**

| Category | Variable | Question/Statement | Measure |
|---|---|---|---|
| Climate change perceptions | Global climate | The global climate is changing. | 5-point Likert scale—from strongly disagree to strongly agree |
| | Global temperature | Average global temperatures are increasing. | 5-point Likert scale |
| | Anthropogenic causes | Human activities such as fossil fuel combustions are an important cause of climate change. | 5-point Likert scale |
| Psychological Distance of Climate Change | | | |
| Temporal Dimension | Today | The effects of climate change are not being felt today. | 5-point Likert scale |
| Spatial dimension | Local agriculture | Climate change does not presents more risk than benefits to agriculture in Puerto Rico. | 5-point Likert scale |
| | Global agriculture | Climate change presents more risks than benefits to agriculture globally. | 5-point Likert scale |
| Social dimension | Farmers | Farmers like me are not likely to be affected negatively by climate change. | 5-point Likert scale |
| | General public | People who are not farmers are likely to be affected negatively by climate change. | 5-point Likert scale |
| Hypothetical or uncertainty dimension | Impact uncertainty | There is scientific uncertainty about the potential impacts of climate change on agriculture. | 5-point Likert scale |
| | Cause uncertainty | There is scientific uncertainty about the causes of climate change. | 5-point Likert scale |
| | Hurricane uncertainty | I am uncertain that the occurrence of strong hurricanes in the Atlantic is related to climate change. | 5-point Likert scale |
| Adaptation perceptions | Motivation to adapt | I feel motivated to change my agricultural practices to prepare for future extreme weather events like Hurricane Maria. | 5-point Likert scale |
| | Perceived self-capacity | I feel that I have the capacity to change my agricultural practices to prepare for future potential extreme weather events like Hurricane Maria. | 5-point Likert scale |
| | Perceived vulnerability | I believe my farm is vulnerable to future extreme weather events like Hurricane Maria. | 5-point Likert scale |
| Main dependent variable | Actual adoption | Which of these agricultural practices and management strategies, if any, might you adopt in the near future to adapt to future extreme events like Hurricane Maria?[a] | Count |
| Experience with extreme weather events | Reported experience | I have faced similar extreme weather events like Hurricane Maria in the past ten years. | 5-point Likert scale |
| | Reported damages | How would you describe the damages, if any, caused by Hurricane Maria to your farm? | 5-point Likert—from no damages to total loss |
| Farmer and farm characteristics* | Age | In which year you were born? | Continuous |
| | Bonafide[b] | Do you participate in the bonafide program of the Puerto Rico Department of Agriculture? | Binary—yes or no |
| | Education level | What is the highest level of education you have completed? | Ordinal—from elementary school to PhD degree |
| | Farm size | How many cuerdas[c] of terrain do you manage in your farm? | Continuous |
| | Gender | What is your gender? | Binary—female or male |
| | Household income | What is your approximate household income, including all far and off-farm income? | Ordinal—from Less than $20,000 to more than $90,000 |

[a]The list of practices can be found in S2 Table. Farmers were asked to report, from a list of predetermined actions, what were the practices currently in use after Maria.

[b]The bonafide program of the Puerto Rico Department of Agriculture provides farmers with agricultural incentives and benefits.

[c]This is the traditional measurement of land use in Puerto Rico. One cuerda is approximately 0.97 acres.

spatial dimensions were assessed with two each. Questions assessing the social and spatial dimensions were intended to separately assess local and distant concerns. Uncertainty was assessed through three statements. Variables were reversed where needed, in order to analyze ascending levels of the psychological distance of climate change; "5" in the scale would represent "psychologically distant". We used Cronbach's alpha to measure internal validity of these constructs in the pilot analysis, demonstrating good internal reliability (alpha = 0.86). Furthermore, a factor analysis using principal components was carried out on these items to further assess consistency. S1 Table shows the pilot data's analysis results, which indicate all items aligned well under one factor (Factor loadings were >.40). This supported our decision to keep a variable scale in the study −Cronbach's alpha for the main study's psychological distance of climate change scale was 0.74, which suggests good internal validity [59].

## Assessing adaptation perceptions and outcomes

Farmers were asked about their perceived self-capacity to change their agricultural practices to adapt to future extreme events, their reported motivation to do so, and their perception around the vulnerability of their farms to the impacts of future extreme weather events. These perceptions were measured through three 5-point Likert scale items. We included motivation to adapt as the intermediary variable between both perceived self-capacity and vulnerability, and actual adoption. Farmers were also asked about the number of practices and management strategies they were currently adopting after Hurricane Maria, in order to adapt to future extreme weather events like Maria. The survey contained a table with a list of 21 practices and management strategies, so farmers could report those they intended to adopt, and those currently adopted. Here we use results for the actual adopted practices (S2 Table), and generate a count variable by summing reported actual adopted practices (a range from 0 to 20). We did not include "exiting farming" in generating this count variable because we aimed to focus on those currently farming (and only six farmers said they had terminated their farming operations). All variables included in the model can be found in Table 1.

## Structural equation model

A structural equation model was built to explore the extent to which psychological distance of climate change relates to farmers' reported experience with similar events to Hurricane Maria and reported degree of damage due to Maria, their motivation to adapt to climate change and their actual adoption of agricultural practices and management strategies. Structural equation models allow quantitative analysis of the direct, indirect, and mediated interactions between variables, and the incorporation of latent constructs [30, 60, 61]. Fig 1 shows the hypothesized model. Two variables assessed experience: "reported past experience" and "reported hurricane damages" (Table 1). Variables for perceived capacity and vulnerability, as well as for motivation to adapt, and actual adoption of agricultural practices were included. The structural equation model was deployed in Stata 15.1, using Maximum Likelihood with bootstrapping (1000 iterations) to control for Type I errors, for non-normal data, and to increase statistical power [30, 62, 63]. The following Goodness-of-fit criteria were used: Standardized Root Mean Square Residual (SRMR), Comparative fit index (CFI), Root mean square error of approximation (RMSEA) [61].

# Results

## Participants' characteristics and the impact of Hurricane María

Farmers had an average age of 54, and farmed an average of 58 *cuerdas* (approximately 56 acres). The great majority of respondents had attended college (67%) (Table 2). Table 3

**Table 2. Mean statistics of control variables.**

| Variable | Scale | Frequency (%) | Mean ± SD | n |
|---|---|---|---|---|
| **Age** | **Continuous** | - | **54.0 ± 13.5** | **391** |
| Bonafide | Yes | 210 (52.8) | - | 398 |
| | No | 188 (47.2) | | |
| Education level | Elementary School | 21 (5.2) | - | 401 |
| | Junior High School | 13 (3.2) | | |
| | Some High School | 15 (3.7) | | |
| | High School Diploma | 82 (20.5) | | |
| | Some College | 42 (10.5) | | |
| | Technical Degree | 25 (6.2) | | |
| | Associate Degree | 41 (10.2) | | |
| | Bachelor's Degree | 109 (27.2) | | |
| | Master's Degree | 46 (11.5) | | |
| | PhD | 7 (1.8) | | |
| Farm size | Continuous | - | 58.1 ± 98.5 | 383 |
| Gender | Female | 55 (14.0) | - | 395 |
| | Male | 340 (86.0) | | |
| Household income | Less than $20,000 | 138 (36.4) | - | 379 |
| | $20,000 - $40,999 | 125 (33.0) | | |
| | $41,000 - $60,999 | 52 (13.7) | | |
| | $61,000 - $80,999 | 36 (9.5) | | |
| | More than $90,000 | 28 (7.4) | | |

contrasts farmers' demographics and farm characteristics with those reported in Puerto Rico's 2017 Agricultural Census [56]. Respondents generally were similar to average census statistics, with the exception of having higher rates of formal higher education and somewhat higher percentage of income from farming. It is important to note that the recent census is not

**Table 3. Demographic variables of our study's surveyed farmers in comparison to USDA 2017 Agricultural Census data for Puerto Rico.**

| Category | Present study | USDA 2017 Census data |
|---|---|---|
| Average farm size | 58.1 *cuerdas* (majority farmed > 20 *cuerdas*) | 59.3 *cuerdas* (majority farmed > 20 *cuerdas*) |
| Years farming | Majority > 10 years | Majority > 10 years |
| Average age | 54 | 61 |
| Gender | Great majority is male | Great majority is male |
| Education | Majority reported High School Diploma or more | Majority reported High School Diploma or less |
| Income from farming | > 50% | < 50% |
| Household income | Majority reported < $20,000 | Majority reported < $20,000 |

Note: Data from the census is for farms' principal operators. In Puerto Rico, this does not necessarily mean that they are the sole owner of the farm. Neither the census or the local Department of Agriculture detail the number of Puerto Rican farmers who participate in the bonafide program; however, in a media report in 2019 [65], according to the Secretary of Agriculture, Carlos Flores, among Puerto Rico's 17,000 farmers, 4,000 are part of the bonafide program; 24% in comparison to the 53% in our study. It is also important to note that the Secretary's comment contrast with the number of farmers provided by the USDA census, which was conducted in 2018, and had a low response rate [64].

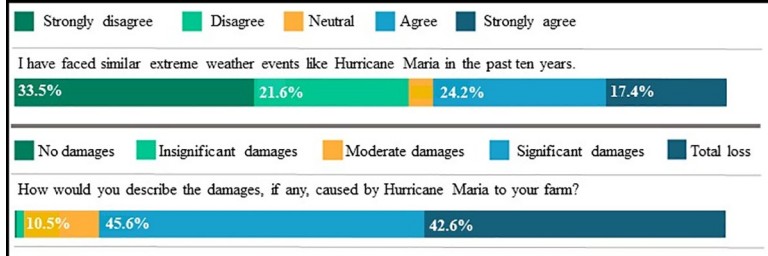

**Fig 2. Farmers' past experience with events similar to Hurricane Maria (top) and reported hurricane damages (bottom).**

comparable to the previous one done in 2012 [56]. Moreover the 2017 census, performed in 2018, had a lower response rate, and USDA reported some difficulty in performing the census given post Hurricane Maria conditions [56, 64]. Farmer respondents were distributed throughout Puerto Rico with 15% in the Arecibo region, 22% in the Caguas region, 19% in the Mayagüez region, 25% in the Ponce region, and 22% in the San Juan region. (Percentages do not sum 100% because some farms extend beyond one municipality that border with another). The majority of farmers experienced damages from Hurricane María; 42.6% reported total loss of their farms, while 45.6% reported significant damages (Fig 2). The majority of farmers (55.1%) disagreed and strongly disagreed that they had faced similar events to Hurricane Maria.

## Climate change and adaptation perceptions

Fig 3 shows results for farmers' perceptions around climate change and adaptation. An overwhelming majority of Puerto Rican farmers believe that the global climate is changing (86% strongly agree and agree), and that it has anthropogenic causes (91.4%). As well, 84.4% of Puerto Rican farmers agreed and strongly agreed that they feel motivated to change their agricultural practices to better prepare for future extreme events. Though farmers perceive their farms to be vulnerable to future extreme weather events (93.5%), they understand themselves capable to adapt to climate change (79.0%).

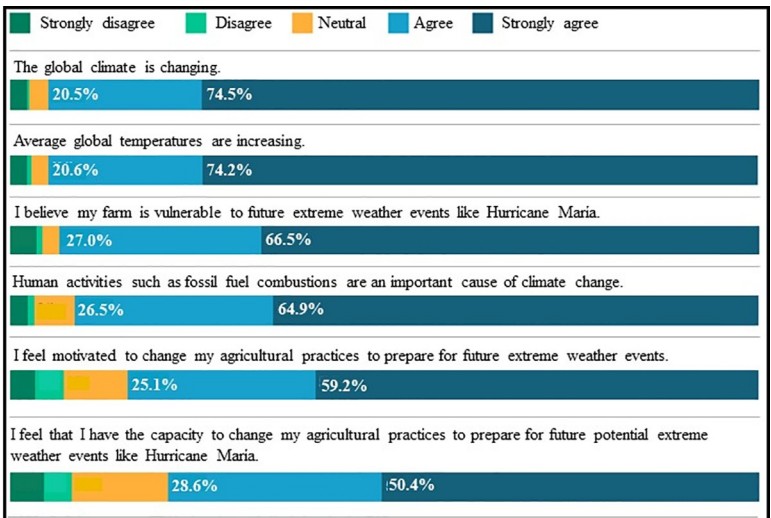

**Fig 3. Climate change perceptions and adaptation perceptions among Puerto Rican farmers.**

**Table 4. Number of adopted agricultural practices and management strategies after Hurricane Maria by Puerto Rican farmers.**

| Number of adopted practices | Frequency (%) |
| --- | --- |
| 0 | 202 (51.0) |
| 1–4 | 98 (25.0) |
| 5–8 | 60 (15.0) |
| 9–12 | 29 (7.0) |
| 13–17 | 7 (1.0) |

## Farmers adoption of agricultural practices and management strategies after Hurricane Maria

Table 4 shows the number of adopted practices and management strategies, and percentage of farmers that adopted those numbers of practices. Almost half of farmers (49%) reported adopting at least one agricultural practice or management strategy to prepare for future events like Maria. On average, farmers adopted 2.5 practices (SD: ± 3.6). S2 Table shows types of agricultural practices and management strategies, their frequencies, and percentage of farmers that adopted them.

## Puerto Rican farmers' psychological distance of climate change

We found that Puerto Rican farmers understand climate change as both a local and global phenomenon, being both psychologically close and psychologically distant across various measures. While we find evidence across all four dimensions of psychological distance for perceived "closeness", we also find that farmers demonstrate awareness for climate change effects beyond Puerto Rico in all dimensions (psychologically far) (Fig 4).

Respondents demonstrated a temporally close relationship to climate change with the majority disagreeing and strongly disagreeing (95.1%) that the effects of climate change are not being felt today. Regarding the spatial dimension, Puerto Rican farmers perceive that climate change will both negatively impact agriculture at both local and global scales (Fig 4). The

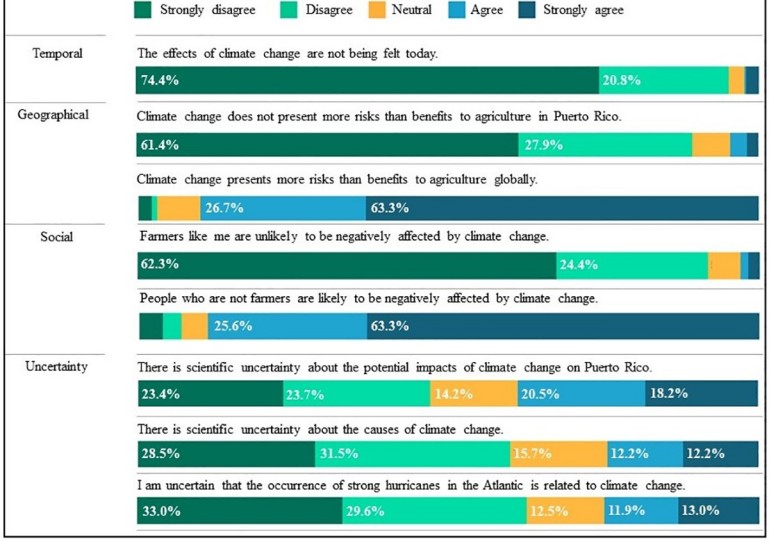

**Fig 4. Results for variables of the psychological distance of climate change.**

majority disagreed and strongly disagreed (89.4%) with the statement, "climate change does not present more risks than benefits to agriculture in Puerto Rico" (e.g. Puerto Rico's agriculture is at risk). Farmers also agreed and strongly agreed (90.0%) with the statement focused on the global scale, "Climate change presents more risks than benefits to agriculture globally". At the social dimension, farmers expressed that climate change will both affect farmers (people like them), and the general public (people not like them). The majority (92%) disagreed and strongly disagreed that farmers like them will likely be affected negatively by climate change (psychologically close); 89% agreed and strongly agreed with the statement, "people who aren't farmers are likely to be negatively affected by climate change" (psychologically far).

We found the greatest variability in the uncertainty or hypothetical dimension of psychological distance of climate change. Thirty nine percent (39%) of respondents agreed and strongly agreed that there is scientific uncertainty about the potential impacts of climate change on Puerto Rico, 14% were neutral, while 47% disagreed and strongly disagreed with the statement. The majority (60%) disagreed and strongly disagreed that there is scientific uncertainty about the causes of climate change; 16% were neutral, and 24% agreed and strongly agreed with the statement. Regarding the statement related to feeling uncertain about the relationship between climate change and occurrence of strong hurricanes in the Atlantic, 25% agreed and strongly agreed, 13% were neutral, and 63% disagreed and strongly disagreed. Furthermore, respondents overwhelmingly agreed that human activities are an important cause of climate change (3.2% disagreed and strongly disagreed, 5.4% were neutral, and 91.4% agreed and strongly agreed).

## Structural equation model

The structural equation model (S3–S5 Tables; Fig 5) demonstrates that four of the seven hypothesized pathways were statistically significant ($p < 0.05$); S3 Table shows results for all variables, including control variables. S4 Table shows results of indirect effects, and S5 Table shows results of total effects. Goodness-of-fit criteria suggests the model is acceptable: RMSEA = 0.062, CFI = 0.809, SRMR = 0.062.

The model shows that reported hurricane damages and reported past experience with similar extreme events to Maria are not significantly predictive of the psychological distance of climate change amongst Puerto Rican farmers, a rejection of H1. Moreover, none of the control variables (age, gender, farm size, bonafide, education, and income) were linked to the psychological distance of climate change (S3 Table). However, both perceived capacity and vulnerability are significant and negatively related to the psychological distance of climate change ($p < 0.05$) (H2). In other words, farmers that perceived climate change as more distant had lower rates of perceived capacity ($b$ = -0.138, $p$ = 0.022), and were less likely to perceive their

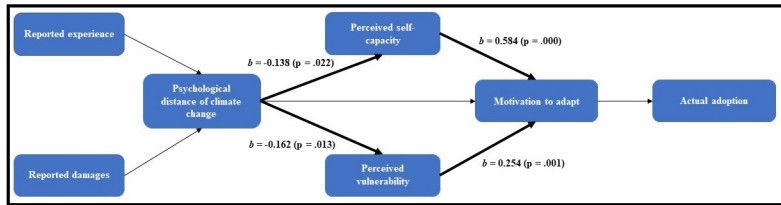

**Fig 5. Structural equation model results for the hypothesized model.** Thick arrows show significance ($p < 0.05$). Standardize coefficients are also shown ($b$). The model controls for farmer and farm characteristics, including age, gender, household income, farm size, education, and bonafide. Goodness-of-fit criteria suggest an overall good fit: RMSEA = 0.062, CFI = 0.809, SRMR = 0.062. Results of indirect effects and total effects are in S4 and S5 Tables, respectively.

farms as vulnerable to future extreme weather events ($b$ = -0.162, p = 0.013). We find no significant direct relationship between psychological distance of climate change and motivation to adapt; however, we did find that it had a significant and negative indirect effect on motivation to adapt ($b$ = -0.176, p = 0.015). This finding suggests that regardless of farmers' reporting higher rates of perceived capacity and vulnerability, motivation to adapt will be negatively impacted if they are psychologically distant.

Both perceived capacity ($b$ = 0.584, p = 0.013) and perceived vulnerability ($b$ = 0.254, p = 0.001) were positively linked to motivation to adapt, suggesting that higher rates of both perceived self-capacity and vulnerability were linked to higher rates of motivation to adapt. Finally, we do not find any significant effect of motivation to adopt on actual adoption of climate adaptation practices and management strategies, a rejection of H3. There were also no indirect effects of other variables on our main dependent variable.

We find no significant relationships between control variables and reported past experience with similar extreme events (S3 Table). Regarding reported hurricane damages, farmers who are participants of the bonafide program of the Puerto Rico Department of Agriculture were more likely ($b$ = 0.136, p = 0.028) to report higher rates of damages to their farms due to Hurricane Maria. Furthermore, education level was negatively linked to reported damages by Hurricane Maria ($b$ = -0.188, p = 0.001), meaning that increasing levels of formal education were linked to reporting lower rates of damages.

## Discussion

This study examined Puerto Rican farmers' actual adoption of agricultural practices and management strategies after Hurricane Maria as a function of psychological distance of climate change, and in relation to perceived capacity, perceived vulnerability, and motivation to adapt. Studies have used psychological distance to climate change to better understand whether this framing can prompt individual mitigation or adaptation actions. Moreover, the trend in the literature is that reduced psychological distance—mainly through experience with extreme weather events—will correlate with action [26, 32–34, 37, 39, 66]. Our results show a different picture. We found that Puerto Rican farmers recognize climate change as a local and global issue, suggesting that climate change is neither psychologically close or far, but instead that farmers have a psychological awareness of climate change's impacts across multiple time frames, geographies, and social constructs. We also found that neither reported experience with past extreme weather events, nor the reported damages effected by Maria were linked to farmers' psychological distance of climate change. Furthermore, farmers overwhelmingly perceived themselves as having the capacity and motivation to adapt to climate change. And though psychological distance did not have a direct effect on motivation to adapt, it did have an indirect effect on it through perceived self-capacity and vulnerability (S4 and S5 Tables). Thus, perceived self-capacity and perceive vulnerability, potentially, mediate psychological distance's effects on reported motivation to adapt. Nevertheless, none of these variables were found to be significantly linked to actual adoption of agricultural practices and management strategies after Hurricane Maria. The majority of farmers did not report adopted practices or management strategies to adapt to future extreme weather events eight months after Hurricane Maria, and motivation to adapt was not significantly related to actual adoption. These results highlight that perceived capacity and motivation to adapt, as well as high levels of climate change belief, are not driving on-farm adaptation behaviors, providing important implications for adaptation research and practice.

Given that farmers actual adoption of agricultural practices and management strategies was not significantly linked to perceived self-capacity, perceived vulnerability, and motivation to

adapt—variables that have been shown to positively affect decision-making around climate change adaptation [9]—future research and practitioners could focus on drivers and barriers at broader levels (e.g. community, institutional, regional, national) to strengthen adaptive capacity. The fact that Puerto Rican farmers report overall perceived capacity and motivation to adapt, and that they understand their farms to be vulnerable, could suggest that farmers may be open to participate in strategies and interventions around climate change adaptation, but may lack other capacities beyond the individual level.

Unlike Spence et al. (2011), we did not find that reported experience with Maria or the psychological distance of climate change were strongly linked to motivation to adapt. The structural equation model (Fig 5) showed that perceiving climate change to be psychologically distant is linked to reporting lower rates of perceived self-capacity. The same was shown for perceived vulnerability to extreme weather events. Farmers that perceived climate change to be far were more likely to report lower rates of perceive farm vulnerability. These findings are consistent with other work [11, 26, 67]. Experiencing climate change impacts or perceiving it to be close may inform how to manage the situation to reduce or avoid impacts (increasing perceive capacity), and inform on the extent of damages those impacts cause (increasing perceived vulnerability) [9, 11, 26, 31, 34]. Nevertheless, farmers' psychological distance of climate change does not explain much of both perceived capacity ($R^2 = 0.019$) and vulnerability ($R^2 = 0.026$). And farmers also showed awareness of climate change impacts both near and far. This could suggest that constant experience with impacts (e.g. cumulative hazards or compound risks) might have become part of the "status quo" [11], and not have a strong influence on reporting that one is capable to adapt or that the system in which one works is vulnerable. This assertion, as well as a more nuanced exploration of perceived capacity and vulnerability, should be further examined.

But perhaps most important, is that we find no relationship between motivation to adapt and actual adoption eight months after Hurricane Maria, suggesting that other barriers to adoption exist, especially in areas constantly affected by extreme weather events. Farmers in Latin America, the Caribbean, and Africa have expressed concern for climate change at different scales, but such perceptions are poorly linked to actual climate change adaptation [40, 68–70]. For example, Harvey and colleagues (2018) found that regardless of perceiving and experiencing climate change risks and impacts, Central American farmers showed low adoption of adaptation practices in response to climate change's impacts, primarily because of low adaptive capacity due to social determinants and structural barriers, such as level of education, household income, access to disaster aid, land tenure, and others [67]. Taking into account the damage caused by Hurricane María and its aftermath, as well as the poor subsequent recovery efforts, our findings suggest that to strengthen farmers' adaptive capacity we must look at other determinants beyond the individual level.

Farmers in Latin America [67, 68], and the Caribbean [70, 71] have been noted to lack institutional structures of support. In Puerto Rico, Perfecto and colleagues (2019) found that social capital and support networks were pivotal for Puerto Rican coffee farmers' recovery after Hurricane María, and that their agroecological practices, such as agroforestry, and other management styles were not sufficient for farms to be resilient and resistant within the catastrophic context of Hurricane Maria [54].

Hurricane María affected all of Puerto Rico, causing 2,975 deaths [72], and decimated the archipelago's agriculture. As such, we argue that reasons and circumstances beyond the self (perceptions) deserve more attention to better understand Puerto Rican farmers' decision-making, and increase their adaptive capacity. This conclusion aligns with a recent review by Wilson and colleagues (2020), which showed that studies on adaptation have taken two roads: one that seeks to understand the role of social and cognitive variables in adaptation behaviors,

and another on how structural factors enact adaptation [73]. It is important to fill in the research gap on how societal and governance structures interact with social and cognitive beliefs in eliciting adaptation behaviors [73].

Our results on overall climate change perceptions, and farmers' psychological distance of climate change, align with a study by the Puerto Rican Department of Natural Resources that state that Puerto Ricans believe in, are aware of, and are concerned about climate change [74]. Puerto Rican farmers, overwhelmingly, understand that the climate is changing. This awareness of climate change as a threat is strong in Latin America and the Caribbean, where farmers and non-farmers are aware of the impacts related to climate change [68, 75, 76]. Surveyed farmers understand that climate change is happening now (temporal), that it will affect farmers and non-farmers (social), and that it will affect local and global agriculture (spatial). These results are in contrast to some existing research from the mainland of the U.S., where climate change belief among farmers is not nearly as high, especially in the perception of its anthropogenic causes [e.g. 77, 78]. Conversely, these results are more aligned with farmer perceptions of climate change from low-income countries, where climate change belief is overwhelmingly high [e.g. 79, 80]. This evidence may suggest that when a threshold for climate change belief is passed (e.g. a large percent of the population acknowledges the issue, potentially achieved by continued exposure to extreme events or a collective social understanding), climate change belief and psychological distance framing may no longer play a significant role in adaptation behaviors. In other words, when enough people recognize climate change as a problem, both locally and globally, it may no longer catalyze action for climate mitigation or adaptation—-other barriers may exist. This should be further studied.

Our respondents were heavily affected by Hurricane María; 42.6% reported total loss, while 45.6% reported significant losses. Given the absolute prevalence of damage across our population, these outcomes may relate to the fact that we find no relationship of reported damage to psychological distance. It is important to underscore that the last hurricane to directly impact Puerto Rico before Maria was 1998's Hurricane Georges. Moreover, Puerto Rico, as well as the Caribbean, have experienced concurrent hazards and been subject to non-landing hurricanes [4, 53, 81]. And given the age span of our participants, assuming they have lived in Puerto Rico, we can say that they have experienced over 50 tropical storms and hurricanes [53, 81]. As such, these results further lend evidence to the idea of a "extreme events" threshold, whereby a certain frequency and intensity of extreme events no longer impacts climate change belief or psychological distance. Nevertheless, "experience" in this study was examined through reported actual damage by the hurricane, as well as reporting experiencing similar events to Maria in the past 10 years. Given that experience has been measured in different ways, resulting in varied results, a future study should inquire deeply about what it means to "experience" an extreme event, and explore this issue longitudinally [82, 83]. Furthermore, looking at experience and climate change perceptions from a qualitative perspective could also elicit a broader understanding about their linkage to experience and adaptation [84].

Summarizing our results and implications, our study shows that the psychological distance of climate change may not be an appropriate framework to understand adaptation behaviors, especially in island settings or other regions where climate change beliefs are overwhelmingly high. Furthermore, as the public sentiment and acknowledgement of climate change grows [85, 86], it indicates that the framework may also need reconsideration in other geographical and social contexts. While farmers in island and disadvantaged areas, who are at the forefront of constant climate change-related impacts, may have already reached a climate change belief saturation, it could be forthcoming in other regions and types of people. These results suggest that, if such things occur, climate change belief will likely not be an important driver of climate change adaptation. Furthermore, that perceived capacity and motivation to adapt were also

unrelated to actual adaptation behaviors post-hurricane further highlight the need to examine structural, political, and other non-individual barriers to adaptation. It is important to understand the degree to which psychological awareness (not distance) of climate change relates to other adaptation perceptions and decision-making, but this should co-exist with further efforts to more clearly examine non-individual barriers to adaptation. Adaptive capacity cannot be reduced to something that is dependent on the individual because then we could be contributing on the assertion that the genesis of vulnerability lies on the individual [87]. Vulnerability to natural hazards is a systemic issue, and farmers' adaptation to climate change, as a decision-making process, is subject to structural interactions that cannot be overlooked.

## Limitations

We note some limitations in this study. First, to the best of our knowledge, there is no previous study looking at Puerto Rican or Caribbean farmers' psychological distance of climate change before and after an extreme weather event. We lack information on the issue before Hurricane Maria or other events. This limits our ability to understand if their psychological awareness is driven by experience. Nonetheless, Extension Service's internal data before Hurricane Maria showed that farmers were aware of climate change's impacts at different scales. Second, farmers surveyed showed higher percentage of income from farming than that reported in census data [53], and we had an overrepresentation of bonafide farmers. Nevertheless, most of the demographic factors of surveyed farmers here align with those of the latest agricultural census for Puerto Rico (Table 3). Third, the variables we used to measure experience might not have capture a nuanced understanding of experience with extreme weather events [82, 83]. Nonetheless, we use variables that have been used in past research regarding the psychological distance of climate change. Furthermore, given that most farmers surveyed for this study receive information from, or are linked to the Extension Service, this might suggest that they have higher access to information and resources, which may affect their understanding of climate change and other related issues. As research in island systems on this topic continues to grow, there is opportunity for broader engagement with farmers, including those not directly integrated with the Extension Service. We also acknowledge that single item measurements for perceived self-capacity, vulnerability, and motivation is a limitation. This is something that can lead to mono-operation bias. Future studies could investigate these concepts through different constructs, quantitatively and qualitatively. Regarding our main dependent variable, we did not evaluate types of practices (e.g. if they are recommended for adaptation) or how other variables than those related to adaptation perceptions had direct effects on actual adoption of agricultural practices. We understand this to be a limitation.

## Conclusion

Farmers' adaptive capacity must be strengthened to decrease their vulnerability to natural hazards. In order to do so, perceptions around adaptation and risk should be considered given that climate change adaptation—a set of decisions at the individual level—is one key action to increase adaptive capacity. Here we examined the extent to which Puerto Rican farmers' adoption of agricultural practices to prepare for future events after Hurricane Maria relates to adaptation perceptions as a function of their psychological distance of climate change. We found that Puerto Rican farmers are psychologically aware of climate change impacts at different levels—local to global. These findings contrast with research in continental and high-income countries that seeks to reduce psychological distance to climate change amongst individuals to prompt mitigation and adaptation actions. And though farmers perceived their farms to be vulnerable to future extreme weather events, they report to perceive themselves capable and

motivated to adapt to climate change. Nevertheless, none of these variables were linked to their actual adoption of agricultural practices and management strategies after Hurricane Maria. As a result, we suggest that factors beyond the individual, including institutional frameworks of support, must be better understood. Research on the psychological distance of climate change should consider that we may have reached a threshold where climate change perceptions are not significant drivers of change. Hence, we suggest that understanding peoples' "psychological awareness of climate change" and its relation to the social-ecological dynamics could provide a more nuanced comprehension of climate change adaptation than focusing on how being "close" or "far" enacts change.

## Supporting information

**S1 Table. Exploratory factor analysis results for pilot study's assessment of the psychological distance of climate change scale.** Factor loadings are shown, as well as the Cronbach's alpha for statements altogether.
(DOCX)

**S2 Table. Reported adoption of agricultural practices and management strategies after Hurricane Maria to prepare for future extreme weather events.** Frequencies and percentage were calculated based upon the 397 farmers that answered this section of the survey.
(DOCX)

**S3 Table. Structural equation model structural results for our hypothesized model.** Standardized coefficients (β), bootstrap standard error (SE), and significance level (*p*) are included.
(DOCX)

**S4 Table. Structural equation model structural results for indirect effects of our hypothesized model.** Standardized coefficients (β), bootstrap standard error (SE), and significance level (*p*) are included.
(DOCX)

**S5 Table. Structural equation model structural results for total effects of our hypothesized model.** Standardized coefficients (β), bootstrap standard error (SE), and significance level (p) are included.
(DOCX)

## Acknowledgments

We acknowledge and thank all the Puerto Rican farmers that participated in this project, and the agricultural agents of the Extension Service of the University of Puerto Rico at Mayagüez, who enumerated the surveys. We are grateful to Dr. Aníbal Ruiz-Lugo, dean of the Extension Service, for leading the logistics and efforts on-site to carry out this project. Much thanks to Maritzabel Morales and the administrative team of the Extension Service for scanning all the surveys and assisting with administrative logistics, and to Olivia Peña for assisting in data entry. We are also thankful to Drs. María del Carmen Rodríguez-Rodríguez, Robinson Rodríguez-Pérez, and Fernando Pérez-Muñoz for assisting in the completion of the project. We are grateful to Dr. Jim Vigoreaux, Agro. Kiria Hurtado, and Carmen González-Toro for proofreading the Spanish version of the survey. We also acknowledge the colleagues who read the first draft of this paper, and provide us with meaningful review comments. Specially, we thank Amy Trubek, Teresa Mares, Diana Hackenberg, Maya Moore, Carolyn Hricko, and Caitlin B. Morgan. We also acknowledge two anonymous reviewers who provided key recommendations to improve this paper.

## Author Contributions

**Conceptualization:** Luis Alexis Rodríguez-Cruz, Meredith T. Niles.

**Data curation:** Luis Alexis Rodríguez-Cruz.

**Formal analysis:** Luis Alexis Rodríguez-Cruz, Meredith T. Niles.

**Funding acquisition:** Meredith T. Niles.

**Investigation:** Luis Alexis Rodríguez-Cruz, Meredith T. Niles.

**Methodology:** Luis Alexis Rodríguez-Cruz, Meredith T. Niles.

**Project administration:** Luis Alexis Rodríguez-Cruz.

**Supervision:** Meredith T. Niles.

**Writing – original draft:** Luis Alexis Rodríguez-Cruz.

**Writing – review & editing:** Meredith T. Niles.

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
