## [Decision Letter · Decision Letter 0]

12 Aug 2020

PONE-D-20-19200

Puerto Rican farmers' psychological awareness of climate change, and adaptation perceptions after Hurricane Maria

PLOS ONE

Dear Dr. Niles,

Thank you for submitting your manuscript to PLOS ONE. After careful consideration, we feel that it has merit but does not fully meet PLOS ONE’s publication criteria as it currently stands. Therefore, we invite you to submit a revised version of the manuscript that addresses the points raised during the review process.

It was reviewed by two experts in the field who have recommended some modification be made prior to acceptance.

I therefore invite you to make these changes and resubmit your manuscript.

Please write a response to reviewers as this will aid review upon resubmission.

I wish you the best of luck with your revisions.

Hope you are keeping safe and well in these difficult times.

We look forward to receiving your revised manuscript.

Kind regards,

Simon Clegg, PhD

Academic Editor

PLOS ONE

"I have read the journal's policy and the authors of this manuscript have the following competing interests: MTN is a member of the board of directors of The Public Library of Science (PLOS). This role has in no way influenced the outcome or development of this work or the peer review process, nor does it alter our adherence to PLOS ONE policies on sharing data and materials."

Reviewers' comments:

Reviewer's Responses to Questions

**Comments to the Author**

1. Is the manuscript technically sound, and do the data support the conclusions?

Reviewer #1: Partly

Reviewer #2: Partly

2. Has the statistical analysis been performed appropriately and rigorously? 

Reviewer #1: Yes

Reviewer #2: No

3. Have the authors made all data underlying the findings in their manuscript fully available?

Reviewer #1: No

Reviewer #2: Yes

4. Is the manuscript presented in an intelligible fashion and written in standard English?

Reviewer #1: Yes

Reviewer #2: Yes

5. Review Comments to the Author

Reviewer #1: This article reports the results of a survey of Puerto Rican farmers aimed at measuring their motivation to adapt to climate-exacerbated hazards (hurricanes) as a function of psychological distance, vulnerability and capacity. They also examine the effect of prior and recent (Maria) experience on distance through a structural equation model. While I found the study to be timely and an interesting contrast to the majority of the literature in this space (e.g., subsistence farmers in developing contexts, big scale commodity farms in developed contexts, etc), I have some issues with the theoretical framing and the conclusions that are drawn given the variability in the data and the way that variables were operationalized. I think perhaps the most interesting story in this data is this idea that psychological distance may not be a critical lens for understanding motivation in some contexts. The dominant narrative is to "bring the risk closer", but there is some evidence that this is not always better, and this data in particular demonstrates that the risk is both near and far, but that it is the perceived capacity to adapt that best explains motivation. I do agree with the author's conclusions that moving beyond individual or psychological variables may be necessary to really understand motivation and adaptation in contexts like this (and others). This is an argument made in a recent review in Nature Climate Change by Wilson et al.

While the analysis seems appropriate and rigorous, I do worry about the fit of the data for a parametric analysis - it is not normally distributed. And given that reality, along with some issues in measurement, it causes me to question some of the conclusions. I do not see that the data has been made available, but perhaps I missed this. And while the writing is fairly strong, there are some grammatical issues that need addressed. My specific line by line comments are in the attached PDF, but I would make these major suggestions.

1) Be more clear in your introduction about your theoretical framework. The intro focuses largely on psychological distance, and to some degree the role of experience, in explaining motivation to adapt. But then other variables are brought into play (vulnerability, capacity). Why are these included? Why are these the likely theoretical drivers of one's motivation to adapt? Why would we expect distance to help explain vulnerability and capacity? What have other studies found about this suite of variables? Again, you focus on experience and distance, but this must be a reason you included these other variables in your model. And given they turn out to be the most important variables, you need a better review up front. While this may be the first study of farmers in small island developing contexts, there are likely other studies that look at actual experience with a hazard and distance...this was not clear from your review. I think you need to think carefully about what background literature is necessary to situate your findings - and it seems that you need to speak to what is known about experience and distance broadly (and in particular actual experience), and then you need to speak to the role of distance, vulnerability and capacity on motivation. I would start by broadly summarizing what might be known in the adaptation literature at large, but then narrowing in on what is known specifically in the context of agriculture (where that exists).

2) Similarly, you kind of gloss over some of the variables in your methods in terms of how they were operationalized. I also have some other issues with your operationalization and then the conclusions you draw. To what extent might experience have played a bigger role if it was a more nuanced measure? See excellent work by Julie Demuth on measuring experience with natural hazards, her work is largely on tornados, but might be worth perusing. I am not convinced that experience does not matter, it just seems that your metric of experience did not matter. One could argue that accumulated experience with hurricanes over time has moved climate change from a distant to a distant and near threat, but that your study does not capture this dynamic process. Greater clarification on your measures is necessary in general - why these items? Are they validated scales? Have they been used elsewhere?

3) Finally, I think the discussion could better situate your findings in the context of other research on behavioral/psychological factors and climate adaptation, as well as other work on farmers and climate beliefs. You mentioned the van Valkengoed and Steg meta-analysis - how do your results match or not with their findings? You mention in your intro that experience was not significant int heir analysis, but what about your other variables? Vulnerability? Capacity? To what extent have these variables been studied in farmer specific studies? Also, there were several studies out of Purdue on Midwestern US farmer climate beliefs. It would be helpful to situate Puerto Rican farmers within this broader context of global agriculture - how are they similar? How are they different? It is not surprising to me that they are taking this threat more seriously, but that the biggest issue might be capacity to act. A more nuanced discussion of this decision process for this group vs. other farmer groups studied would be useful. Behavior change is often a function of both motivation and ability - Puerto Rican farmers seem to have the motivation (concern, perceived personal risk, etc) but not the ability. While other US farmers (like in big Midwest commodity ag) may lack that initial motivation or concern.

Reviewer #2: This paper provided a comprehensive understanding about the relationships among Puerto Rican farmers’ psychological distance of climate change, their experience of the category 4 Hurricane Maria and their perceptions of climate change. Then the paper explored further the association of these variables to farmers’ motivation to climate change adaptation.

It is worth understanding the roles of extreme weather events experience and perceived psychological distance in farmers’ decision making around climate change adaptation. Besides, this paper supplement a research in an island-archipelago setting to a more comprehensive view of perceptions and adaptation of climate change, which is valuable for future researches and practices.

However, there are some issues (such as the definition of psychological distance and its analysis in the models) that need to be better considered according to the survey and objective design. Also, the results need to be better interpreted for a more concrete conclusion. Below are my comments:

1. Main: about the psychological distance

In the Theoretical framework section (P6), it gave the explanation of psychological distance with the four components. Then according to the Objective, research questions, and hypotheses section (P11), it’s mainly on how the degree of psychological distance (how far or how close) linked to the extreme experience or damage, and how it associates with the cognitive perceptions and motivation of adaptation. However, in Methodology on Assessing the psychological distance (P13, line 285) and the survey (Table 1), it redefined “psychologically far” and “psychologically close” for each item. There are some questions about it:

1）Line 285, “Strongly agree” and “agree” answers for dimensions’ statements were combined as “psychologically far” — How about the reversed items? Usually, negatively worded statements were used as reversed items for controlling the response bias with reverse scoring. But here, it seems only the meaning of local or global were used as the reversed items, instead of negative statements. Need to be careful about this for the further analysis.

2) According to such redefinition and analysis, it has regarded close and far as a one-dimensional variable (either close or far) instead of two-dimensional (how close and how far in those four components at the same time). Nevertheless, the design of the survey items divided close and far into two items, which made it a bit confusing. The results also showed the farmers could perceive both close and far. So maybe there is also a need to consider the close or far into two dimensions for analysis.

It could change to the exploration of the relationship between how close they perceived (the degree of close) in four components and their experience or beliefs. Further, how far they perceived (the degree of far) in those components need to be separated for the two items (General public and Global agriculture).

3) Based on above, if focusing on the degree of psychological distance perceived, maybe keeping 5-point Likert scale coding for analyses would be suitable than 3-points.

4) on Line 284, it showed the Reliability Analysis with Cronbach’s alpha value for the all items in psychological distance scale, could you also provide the Cronbach’s alpha value for each component (more than one item) to see the internal consistency？

5) Besides Reliability Analysis, could you also provide the results of validity analysis, such as factor analysis? This could help to see whether the analysis of psychological distance scale could provide the support for the two dimensions on close and far, or four dimension on theoretical components. Then according to the factor analysis, it could be better consider whether still regard psychological distance as one variable in the models.

Based on the above, I suggest that the authors better consider and interpret the psychological distance and related results. Would they be confident of their final results that “the experiences were not linked to their psychological distance of climate change, and these did not relate to their motivation to adapt”, when using the current ways on the definition and analyses.

2. Main: about the experience (with events and damages)

1）Line 184-185, it seems the direct experience in this paper means the reported degree of damage. But how does one distinguish the reported experience from reported damage level? People could perceive the events and reported the experience through the damage experience. So, it is hard to say that those researches only used data on reported experience that were not related to direct experience, such as Spence’s work cited in this paper.

2）Line 201, it said the paper would assess the extent to which the experience with Maria (both reported experience with other extreme events…). And in Objective section, the Objective 2 and H1 all mentioned the experience with Maria. While based on the only one item for the reported experience in table 1, is the statement framing mainly for the experience with other extreme weather events except Hurricane Maria or for the experience with Hurricane Maria?

3）If the reported experience is focusing on the experience with other extreme weather events except Hurricane Maria, could you provide more introduction on the extreme weather events happened in this area before Maria?

4）Besides, it is needed to clarify the interpretation framing with the experience related results, if the experience from the survey is not experiencing Maria, e.g. in Abstract “Reported experience and direct damages by the hurricane were not linked to their psychological distance”.

5）Line 180, it mentioned the extreme events may have become “more normal” and this view was also mentioned in discussion to explain the weak role of experience in the model. It also could be better if you could provide more background information on the historical extreme events the people there may experienced.

Other comments:

3. line 267, could you clarify the ways that the surveys were randomly administered to farmers in Data gathering section?

4. Adding the Cronbach’s alpha value into the table 1 if it is available.

5. Adding the citation into table 1 if there are some items referring to other researches.

6. Could combine table 2 and table 3 into one comprehensive table, then could consider to put it into supplementary sheet if there is more table/figures than requested.

6. PLOS authors have the option to publish the peer review history of their article (what does this mean?). If published, this will include your full peer review and any attached files.

Reviewer #1: No

Reviewer #2: No

---

## [Author Response · Author response to Decision Letter 0]

25 Sep 2020

Response to Reviewers

Dear Editor and Reviewers,

 We thank you for taking the time and energy to review our manuscript. Your thoughtful comments and recommendations have been valuable in improving our manuscript’s clarity and rigor. Below you can find our responses to specific reviewer comments in bold. At the end, we also include the updated Competing Interests section, as requested in the PLOS ONE Decision email. 

PONE-D-20-19200

Reviewer #1: This article reports the results of a survey of Puerto Rican farmers aimed at measuring their motivation to adapt to climate-exacerbated hazards (hurricanes) as a function of psychological distance, vulnerability and capacity. They also examine the effect of prior and recent (Maria) experience on distance through a structural equation model. While I found the study to be timely and an interesting contrast to the majority of the literature in this space (e.g., subsistence farmers in developing contexts, big scale commodity farms in developed contexts, etc), I have some issues with the theoretical framing and the conclusions that are drawn given the variability in the data and the way that variables were operationalized. I think perhaps the most interesting story in this data is this idea that psychological distance may not be a critical lens for understanding motivation in some contexts. The dominant narrative is to "bring the risk closer", but there is some evidence that this is not always better, and this data in particular demonstrates that the risk is both near and far, but that it is the perceived capacity to adapt that best explains motivation. I do agree with the author's conclusions that moving beyond individual or psychological variables may be necessary to really understand motivation and adaptation in contexts like this (and others). This is an argument made in a recent review in Nature Climate Change by Wilson et al.

While the analysis seems appropriate and rigorous, I do worry about the fit of the data for a parametric analysis - it is not normally distributed. And given that reality, along with some issues in measurement, it causes me to question some of the conclusions. I do not see that the data has been made available, but perhaps I missed this. And while the writing is fairly strong, there are some grammatical issues that need addressed. My specific line by line comments are in the attached PDF, but I would make these major suggestions.

1) Be more clear in your introduction about your theoretical framework. The intro focuses largely on psychological distance, and to some degree the role of experience, in explaining motivation to adapt. But then other variables are brought into play (vulnerability, capacity). Why are these included? Why are these the likely theoretical drivers of one's motivation to adapt? Why would we expect distance to help explain vulnerability and capacity? What have other studies found about this suite of variables? Again, you focus on experience and distance, but this must be a reason you included these other variables in your model. And given they turn out to be the most important variables, you need a better review up front. While this may be the first study of farmers in small island developing contexts, there are likely other studies that look at actual experience with a hazard and distance...this was not clear from your review. I think you need to think carefully about what background literature is necessary to situate your findings - and it seems that you need to speak to what is known about experience and distance broadly (and in particular actual experience), and then you need to speak to the role of distance, vulnerability and capacity on motivation. I would start by broadly summarizing what might be known in the adaptation literature at large, but then narrowing in on what is known specifically in the context of agriculture (where that exists).

Response: We have restructured our introduction, and have broadened specific lines, as well as in the literature review section, to be more specific about the extent to which experience and psychological distance relate, and about why we included the variables you mentioned. We focused on literature that used the psychological distance of climate change (PDCC) as its framework to understand behavior or literature that evaluated PDCC after an event. The main argument of such literature is that reducing psychological distance will prompt mitigation or adaptation behaviors. Here, instead of evaluating adoption of practices, for example, we assessed the extent to which PDCC is linked to reported levels of perceived capacity, vulnerability, and motivation to adapt, which are variables important in adaptive capacity research from a social and psychological perspective. We argued that in order to understand adaptation, which reflects adaptive capacity, we must understand social and cognitive drivers. Thus, we used PDCC to understand how Puerto Rican farmers perceive climate change after the brunt event of Hurricane Maria, how that then that was linked to sociodemographic and reported experience variables, and how PDCC was linked to other adaptation perceptions. Hence, our main goal in this paper was to assess variables that to the best of our understanding have not been studied in Puerto Rico. In the reviewed discussion we underscored that future studies should inquire more deeply about what explains farmers’ perceived capacity, vulnerability, and motivation to adapt from different angles, given that those variables have been shown to be pivotal in adaptation research. It is important to reiterate that this paper is about understanding farmers’ PDCC after a category 4 hurricane.

In terms of your comments regarding the discussion, we appreciate that you highlighted Wilson et al. (2020). We were not aware of that review paper, and we think that their argument aligns we our discussion. We included it in our discussion, thank you. 

Regarding your comments on the structural equation model (SEM), we have used bootstrapping (n=1,000), a documented method to control for a number of issues in structural equation models, including non-normal data. We have clarified in the methods section that using Bootstrap in SEM is recommended for non-normal data. We also intend to share the non-identified data in a repository after the acceptance of the paper.

2) Similarly, you kind of gloss over some of the variables in your methods in terms of how they were operationalized. I also have some other issues with your operationalization and then the conclusions you draw. To what extent might experience have played a bigger role if it was a more nuanced measure? See excellent work by Julie Demuth on measuring experience with natural hazards, her work is largely on tornados, but might be worth perusing. I am not convinced that experience does not matter, it just seems that your metric of experience did not matter. One could argue that accumulated experience with hurricanes over time has moved climate change from a distant to a distant and near threat, but that your study does not capture this dynamic process. Greater clarification on your measures is necessary in general - why these items? Are they validated scales? Have they been used elsewhere?

Response: Thank you for your comments on experience and for bringing Demuth’s work to our attention. We clarified in the paper about our use of the experience variables, both reported hurricane damages and reported experience with similar events. The word “reported” was added to be specific about the variables. Furthermore, in the reviewed discussion, we expanded on “experience” to capture your comments and recommendations. It was also included in the limitations section. Nevertheless, it is important to note that in in the literature review section (Research on the psychological distance of climate change), we highlighted the work by Zanocco et al. (2018) to inform why used reported damages as one measure of experience. Other PDCC studies have used the single reported experience scale, and other studies have been done in an area after an extreme event without asking about that experience. We clarified in the methods about the studies that informed our survey instrument, and how variables have been used in previous studies. Nevertheless, as Demuth shows, the use of different experience measures is a current barrier in existing studies. We recommended the further investigation of what it means to experience an extreme event in Puerto Rico, and we also added historical information on landing and non-landing hurricanes and tropical storms in Puerto Rico. It is important to note that our questions were framed to capture participants’ opinions and perceptions around Hurricane Maria. Puerto Rico was still recovering when the survey was carried out. We also included a section to explain the assessment of social and cognitive variables, and we included more information on how the PDCC scale was built. Finally, we included supplementary material on the pilot’s PDCC analysis, which in demonstrating reliable validity, was another factor in our design. 

3) Finally, I think the discussion could better situate your findings in the context of other research on behavioral/psychological factors and climate adaptation, as well as other work on farmers and climate beliefs. You mentioned the van Valkengoed and Steg meta-analysis - how do your results match or not with their findings? You mention in your intro that experience was not significant int heir analysis, but what about your other variables? Vulnerability? Capacity? To what extent have these variables been studied in farmer specific studies? Also, there were several studies out of Purdue on Midwestern US farmer climate beliefs. It would be helpful to situate Puerto Rican farmers within this broader context of global agriculture - how are they similar? How are they different? It is not surprising to me that they are taking this threat more seriously, but that the biggest issue might be capacity to act. A more nuanced discussion of this decision process for this group vs. other farmer groups studied would be useful. Behavior change is often a function of both motivation and ability - Puerto Rican farmers seem to have the motivation (concern, perceived personal risk, etc) but not the ability. While other US farmers (like in big Midwest commodity ag) may lack that initial motivation or concern.

Response: Thank you for your comments and recommendations. We have expanded our discussion. We think that understanding the decision-making process around adaptation should be further looked into in future studies. We explain in our discussion why this is important, and how our results support those recommendations. 

Reviewer #2: This paper provided a comprehensive understanding about the relationships among Puerto Rican farmers’ psychological distance of climate change, their experience of the category 4 Hurricane Maria and their perceptions of climate change. Then the paper explored further the association of these variables to farmers’ motivation to climate change adaptation.

It is worth understanding the roles of extreme weather events experience and perceived psychological distance in farmers’ decision making around climate change adaptation. Besides, this paper supplement a research in an island-archipelago setting to a more comprehensive view of perceptions and adaptation of climate change, which is valuable for future researches and practices.

However, there are some issues (such as the definition of psychological distance and its analysis in the models) that need to be better considered according to the survey and objective design. Also, the results need to be better interpreted for a more concrete conclusion. Below are my comments:

1. Main: about the psychological distance

In the Theoretical framework section (P6), it gave the explanation of psychological distance with the four components. Then according to the Objective, research questions, and hypotheses section (P11), it’s mainly on how the degree of psychological distance (how far or how close) linked to the extreme experience or damage, and how it associates with the cognitive perceptions and motivation of adaptation. However, in Methodology on Assessing the psychological distance (P13, line 285) and the survey (Table 1), it redefined “psychologically far” and “psychologically close” for each item. There are some questions about it:

1）Line 285, “Strongly agree” and “agree” answers for dimensions’ statements were combined as “psychologically far” — How about the reversed items? Usually, negatively worded statements were used as reversed items for controlling the response bias with reverse scoring. But here, it seems only the meaning of local or global were used as the reversed items, instead of negative statements. Need to be careful about this for the further analysis.

Response: Thank you for the opportunity to further clarify our method, and we apologize for the initial confusion. The psychological distance of climate change (PDCC) scale was developed from eight, separate 5-point Likert scale items. Your reference above is related to our presentation of results in figure 4. Statements were not combined for statistical analysis. Given that the scale measured ascending PDCC (e.g. the higher the number, the more distant the farmer perceives climate change), we presented the results as “psychologically far” for the combination of “agree” and “strongly agree” responses, and “psychologically close” for “disagree” and “strongly disagree responses”. We have decided to eliminate that way of showing results to prevent confusion, and we clarified how we analyzed results in the Methods section. 

2) According to such redefinition and analysis, it has regarded close and far as a one-dimensional variable (either close or far) instead of two-dimensional (how close and how far in those four components at the same time). Nevertheless, the design of the survey items divided close and far into two items, which made it a bit confusing. The results also showed the farmers could perceive both close and far. So maybe there is also a need to consider the close or far into two dimensions for analysis.

It could change to the exploration of the relationship between how close they perceived (the degree of close) in four components and their experience or beliefs. Further, how far they perceived (the degree of far) in those components need to be separated for the two items (General public and Global agriculture).

Response: All the items were evaluated as 5-point Likert scales. To ideally control for “straight-lining” in a survey, we intentionally had some questions worded oppositely so that we could conduct data quality checks. Some questions were reversed so that responses were shown in ascending levels of PDCC. In terms of interpretation, we interpreted “agree” and “strongly agree” to be “psychologically far”, and “agree” and “strongly agree” to be psychologically close. We have clarified in the manuscript how we analyzed and interpret these items and the PDCC scale.

3) Based on above, if focusing on the degree of psychological distance perceived, maybe keeping 5-point Likert scale coding for analyses would be suitable than 3-points.

Response: We clarified in the manuscript that they were interpreted and statistically analyzed as 5-point Likert scales.

4) on Line 284, it showed the Reliability Analysis with Cronbach’s alpha value for the all items in psychological distance scale, could you also provide the Cronbach’s alpha value for each component (more than one item) to see the internal consistency?

Response: We included in the Methods section more information on how we created the scale from the pilot study’s data. We report reliability analysis with Cronbach’s alpha and factor analysis results. Alpha is recommended for scales with three or more items. We expanded in our Methods section about what informed our variable use. 

5) Besides Reliability Analysis, could you also provide the results of validity analysis, such as factor analysis? This could help to see whether the analysis of psychological distance scale could provide the support for the two dimensions on close and far, or four dimension on theoretical components. Then according to the factor analysis, it could be better consider whether still regard psychological distance as one variable in the models.

Based on the above, I suggest that the authors better consider and interpret the psychological distance and related results. Would they be confident of their final results that “the experiences were not linked to their psychological distance of climate change, and these did not relate to their motivation to adapt”, when using the current ways on the definition and analyses.

Response: Thank you for your comments and recommendations. We better explained what we meant by “experience” in the paper, and we included historical records of landing and non-landing hurricanes in Puerto Rico to our discussion. We also performed a reliability analysis and factor analysis with the pilot data, which informed our PDCC analysis. We expanded on Methods, and included the new supplementary materials. 

2. Main: about the experience (with events and damages)

1）Line 184-185, it seems the direct experience in this paper means the reported degree of damage. But how does one distinguish the reported experience from reported damage level? People could perceive the events and reported the experience through the damage experience. So, it is hard to say that those researches only used data on reported experience that were not related to direct experience, such as Spence’s work cited in this paper.

Response: In our revision, we clarified on what we meant by experience, how we assessed it, and what can be done in the future to further understand what it means to experience an extreme weather event in Puerto Rico. We had two variables, one that asked about reported experience with similar events in the past 10 years, and one that asked about farm-level damaged caused by Maria. It is important to note that Hurricane Maria affected all of Puerto Rico. It has been the most devastating storm to hit Puerto Rico in 89 years. People were still recovering when the study was carried out. We also included historical records in the discussion to further expand on experience and PDCC.

2）Line 201, it said the paper would assess the extent to which the experience with Maria (both reported experience with other extreme events…). And in Objective section, the Objective 2 and H1 all mentioned the experience with Maria. While based on the only one item for the reported experience in table 1, is the statement framing mainly for the experience with other extreme weather events except Hurricane Maria or for the experience with Hurricane Maria?

Response: Thank you, we clarified the variables in the objectives, methods, and overall discussion. 

3）If the reported experience is focusing on the experience with other extreme weather events except Hurricane Maria, could you provide more introduction on the extreme weather events happened in this area before Maria?

Response: We included data on historical records regarding hurricanes in the discussion.

4）Besides, it is needed to clarify the interpretation framing with the experience related results, if the experience from the survey is not experiencing Maria, e.g. in Abstract “Reported experience and direct damages by the hurricane were not linked to their psychological distance”.

Response: We clarified how we understand and assessed experience. 

5）Line 180, it mentioned the extreme events may have become “more normal” and this view was also mentioned in discussion to explain the weak role of experience in the model. It also could be better if you could provide more background information on the historical extreme events the people there may experienced.

Response: We have included these suggestions.

Other comments:

3. line 267, could you clarify the ways that the surveys were randomly administered to farmers in Data gathering section?

Response: We expanded on this in the methods section.

4. Adding the Cronbach’s alpha value into the table 1 if it is available.

Response: Cronbach’s alpha for the main study’s scale is included in the text of the Assessing the psychological distance of climate change section in Methods.

5. Adding the citation into table 1 if there are some items referring to other researches.

Response: Thank you for the recommendation. We added the citations in text.

6. Could combine table 2 and table 3 into one comprehensive table, then could consider to put it into supplementary sheet if there is more table/figures than requested.

Response: We have decided to keep the tables separate. Thank you for the recommendation. 

Here is our update Competing Interests section: 

“I have read the journal's policy and the authors of this manuscript have the following competing interests: MTN is a member of the board of directors of The Public Library of Science (PLOS). This role has in no way influenced the outcome or development of this work or the peer review process, nor does it alter our adherence to PLOS ONE policies on sharing data and materials. This does not alter our adherence to PLOS ONE policies on sharing data and materials.”

---

## [Decision Letter · Decision Letter 1]

20 Oct 2020

PONE-D-20-19200R1

Puerto Rican farmers' psychological awareness of climate change, and adaptation perceptions after Hurricane Maria

PLOS ONE

Dear Dr. Rodriguez-Cruz

Thank you for submitting your manuscript to PLOS ONE. After careful consideration, we feel that it has merit but does not fully meet PLOS ONE’s publication criteria as it currently stands. Therefore, we invite you to submit a revised version of the manuscript that addresses the points raised during the review process.

Many thanks for resubmitting your manuscript to PLOS One

It was reviewed by the same reviewers as last time, and they have recommended some modifications be made prior to acceptance.

If you could write a brief response to reviewers that will expedite review when resubmitted

I wish you the best of luck with your changes

Hope you are keeping safe and well in these difficult times

Thanks

Simon

We look forward to receiving your revised manuscript.

Kind regards,

Simon Clegg, PhD

Academic Editor

PLOS ONE

Reviewers' comments:

Reviewer's Responses to Questions

**Comments to the Author**

1. If the authors have adequately addressed your comments raised in a previous round of review and you feel that this manuscript is now acceptable for publication, you may indicate that here to bypass the “Comments to the Author” section, enter your conflict of interest statement in the “Confidential to Editor” section, and submit your "Accept" recommendation.

Reviewer #1: (No Response)

Reviewer #2: All comments have been addressed

2. Is the manuscript technically sound, and do the data support the conclusions?

Reviewer #1: No

Reviewer #2: Yes

3. Has the statistical analysis been performed appropriately and rigorously? 

Reviewer #1: No

Reviewer #2: Yes

4. Have the authors made all data underlying the findings in their manuscript fully available?

Reviewer #1: No

Reviewer #2: Yes

5. Is the manuscript presented in an intelligible fashion and written in standard English?

Reviewer #1: Yes

Reviewer #2: Yes

6. Review Comments to the Author

Reviewer #1: Unfortunately, I do not feel that my prior comments were sufficiently addressed. This paper still suffers from a lack of proper set-up in the introduction to support the analyses and related discussion. The paper is framed as a study of experience, psychological distance and adaptation. However, the study then diverges to include other key beliefs that may influence one's motivation to adapt. Yet, the inclusion of these beliefs in the SEM, and the theoretical framework guiding their inclusion (and placement in the proposed structure) is not addressed. For example, why would we expect distance to influence vulnerability and self-capacity? I can intuit the former, but not the latter. Theoretically, the perceived distance of climate change and perceived vulnerability would work in parallel with self-capacity to influence adaptation decisions. e.g., One can be concerned about local impacts and be motivated to act, but not able to act due to limited capacity. I don't think the paper properly deals with these constructs - the issues begin the measurement of these ideas with single items, but extend to the proper theoretical hypotheses for their relationships. The authors in fact make this argument in the discussion that perhaps the bigger issue in Puerto Rico are the structural barriers to action, not a lack of appropriate concern about climate change, but then why do the respondents report such high self capacity? If you are intending to differentiate between motivation and intention, then be more clear about this. You didn't measure intention to adapt or actual adaptation, you measure "motivation" - but I think most of the behavioral literature would argue that motivation can be high, but ability low, and then there is a failure to act.

I would suggest a few paths forward for the authors.

1) Rethink your conceptual model. If the strength of your study is the experience, distance and motivation piece, then stick to that. Your measures of vulnerability and self-capacity are not strong (single item) and their position in the proposed model is not well defined or defended. In fact, their position (self-capacity in particular) runs contrary to much of your conclusion around the importance of structural barriers. If structural barriers are so key? Then why are your respondents reporting such high capacity? If you focus on distance and experience, which your intro sets up sufficiently, then perhaps split your distance items in your SEM to look at separate paths through each dimension. Or perhaps through the "near" vs. "far" dimensions. If you measure is capturing a continuum from near to far and respondents score strongly on both, then combining those items will wash out any potential effect. Separating them out would allow you to test if a particular perspective is driving their motivation - you may find that experience does drive the "near" dimension, and that the near dimension does drive motivation. The paper almost reads like you intended this to be a study on distance and experience, but then there were no effects so you threw in a few other relevant items to have a story to tell. I don't disagree with your story and conclusions in theory, but I don't think your data support that story.

2) Think carefully about your language and terminology throughout the paper - you talk about motivation to adapt, adaptive capacity, self capacity, cognitive beliefs, context, structure, adaptation beliefs, etc. etc. It isn't always clear what you are talking about and if you are using terms interchangeably (e.g., motivation to adapt vs. adaptive capacity) or if you are using certain phrases to refer to sets of your constructs (e.g., adaptation beliefs for everything vs. cognitive beliefs for three of your measures). Choose one set of terms for the idea in the paper and define them clearly at first mention, then use them consistently throughout. And be sure that your terminology reflects best practice and what will be most intuitive to those familiar with this literature.

My specific line by line comments are attached in the PDF.

Reviewer #2: (No Response)

7. PLOS authors have the option to publish the peer review history of their article (what does this mean?). If published, this will include your full peer review and any attached files.

Reviewer #1: No

Reviewer #2: No

---

## [Author Response · Author response to Decision Letter 1]

7 Dec 2020

Dear Reviewer, here are the comments to both of your comments:

1)

Response: Thank you for your recommendations. We agree that single-item measurements are a limitation, which we include in our limitations section. To address your concerns, we have expanded sections of the paper, especially those areas that focused on perceived self-capacity to better set up the context and framing for their inclusion. Your comments were helpful to improve consistency regarding concepts used, especially in motivating us to better clarify on the differences between “adaptive capacity” and “climate change adaptation”. It is important to note that this paper was based upon Spence et al. (2011), where they look at United Kingdom residents’ climate change mitigation support in light of their experience with floods, in function of experience driving concern for climate change, uncertainty, perceived capacity, and perceived vulnerability. Their study has been the baseline for studies using the psychological distance of climate change as a framework. Our model is built upon their findings, including the similar constructs. We hope that the additions of our clarifications, including the focus of Spence as a guiding framework, helps further clarify our work. However, in reviewing your feedback, we also considered our model further, specifically as it related to your comment about why if people had so much perceived capacity is there a problem? Ultimately, our model was only measuring motivation to adopt, not actual adoption, and literature suggests that such constructs are critically different. Thus, in this revision, we have added a new dependent variable: actual adoption of agricultural practices and management strategies after Maria. The addition of this variable helps complete a more nuanced understanding, and connection between capacity, motivation and actual behavioral change. As a result of adding this construct, we believe that a more complete story is told with these results- while perceived capacity is high, and motivation to adapt is high, actual adoption of adaptation practices is not related to motivation to adapt. This clearly shows the gap and barriers between motivation and perceived capacity and actual behaviors, which are hindered potentially by other factors, including structural factors, that we mention in our discussion. 

2)

Response: Thank you for providing this additional feedback, and your pdf comments on the concepts used. We have done a careful review of our terms and reviewed them in our revisions. We better defined the concepts, and made revisions for them to be constant in the manuscript.

---

## [Editor Report · Decision Letter 2]

11 Dec 2020

Awareness of climate change's risks and motivation to adapt are not enough to drive action: A look of Puerto Rican farmers after Hurricane Maria

PONE-D-20-19200R2

Dear Dr. Rodriguez-Cruz

We’re pleased to inform you that your manuscript has been judged scientifically suitable for publication and will be formally accepted for publication once it meets all outstanding technical requirements.

Kind regards,

Simon Clegg, PhD

Academic Editor

PLOS ONE

Additional Editor Comments

Many thanks for resubmitting your manuscript to PLOS One

As all the comments have been addressed and the manuscript reads well, I have recommended it for publication

You should hear from the Editorial Office soon

It was a pleasure working with you, and I wish you all the best for your future research

Hope you are keeping safe and well in these difficult times

Thanks

Simon

---

## [Editor Report · Acceptance letter]

7 Jan 2021

PONE-D-20-19200R2 

Awareness of climate change's impacts and motivation to adapt are not enough to drive action: A look of Puerto Rican farmers after Hurricane Maria 

Dear Dr. Rodríguez-Cruz:

I'm pleased to inform you that your manuscript has been deemed suitable for publication in PLOS ONE. Congratulations! Your manuscript is now with our production department. 

Kind regards, 

on behalf of

Dr. Simon Clegg 

Academic Editor

PLOS ONE